# Foraging, Fear and Behavioral Variation in a Traplining Hummingbird

**DOI:** 10.3390/ani13121997

**Published:** 2023-06-15

**Authors:** Katarzyna Wojczulanis-Jakubas, Marcelo Araya-Salas

**Affiliations:** 1Department of Vertebrate Ecology and Zoology, University of Gdansk, 80-308 Gdansk, Poland; 2Centro de Investigación en Neurociencias, University of Costa Rica, San José 11501-2060, Costa Rica; marcelo.araya@ucr.ac.cr

**Keywords:** risk allocation hypothesis, risk avoidance, exploratory behavior, arousal, repeatability

## Abstract

**Simple Summary:**

Animals differ in their foraging efficiency and it is not clear what drives this variation. We examined how the foraging efficiency of long-billed hermit hummingbirds changes with regard to three behavioral traits: (a) exploration, (b) risk avoidance and (c) arousal in conditions at two different levels of perceived risk (low and high). We found that foraging efficiency was lower in high-risk conditions, but behavioral traits explained the additional variation in foraging efficiency in a condition-dependent manner. More explorative individuals had higher foraging efficiency in low-risk conditions, but the opposite was the case in high-risk conditions. Regardless of the conditions, foraging efficiency increased with bird arousal and decreased if they were more cautious (higher risk avoidance). Our findings highlight the importance of taking into account additional behavioral dimensions to better understand the foraging strategies of individuals.

**Abstract:**

Traditionally, foraging behavior has been explained as the response to a trade-off between energetic gain from feeding resources and potential costs from concomitant risks. However, an increasing number of studies has shown that this view fails to explain an important fraction of the variation in foraging across a variety of taxa. One potential mechanism that may account for this variation is that various behavioral traits associated with foraging may have different fitness consequences, which may depend on the environmental context. Here, we explored this mechanism by evaluating the foraging efficiency of long-billed hermit hummingbirds (*Phaethornis longirostris)* with regard to three behavioral traits: (a) exploration (number of feeders used during the foraging visit), (b) risk avoidance (latency to start feeding) and (c) arousal (amount of movements during the foraging visit) in conditions at two different levels of perceived risk (low—control and high—experimental, with a threatening bullet ant model). Foraging efficiency decreased in response to threatening conditions. However, behavioral traits explained additional variation in foraging efficiency in a condition-dependent manner. More exploration was associated with a higher foraging efficiency under control conditions, but this was reversed when exposed to a threat. Regardless of the conditions, arousal was positively associated with foraging efficiency, while risk avoidance was negatively related. Importantly, exploratory behavior and risk avoidance were quite repeatable behaviors, suggesting that they may be related to the intrinsic traits of individuals. Our findings highlight the importance of taking into account additional behavioral dimensions to better understand the foraging strategies of individuals.

## 1. Introduction

A variety of ecological factors have been identified as major determinants in shaping the foraging strategies of animals (i.e., resource exploitation). Of those, the most commonly evoked are the amount and distribution of available food resources [1,2] and animal motivation (both in the sense of the marginal value theorem [3] and/or body condition [4]), but predation pressure is also frequently considered [5,6]. The intensity of prey vigilance increases with the level of risk predation (probability of predator presence), affecting its foraging efficiency. As imposed according to the risk allocation hypothesis, the prey allocates time for foraging inversely proportional to the predation pressure [5,6,7]. Thus, with given food resources or a predation landscape, a fixed foraging strategy is expected to evolve [8]. However, an increasing number of studies demonstrates a high variation in foraging strategies [9,10,11], which is hard to explain using only food- and predation-based arguments [4].

Variation in the foraging strategies of individuals remains an intriguing topic [1,12,13,14,15,16]. Existing evidence demonstrates not only the variation per se, but also the consistent inter-individual differences in the average level of a behavior displayed across a range of contexts [17], and in response to environmental variation [12,16], a species/population and individuals usually represent a full continuum of a given behavioral display, with two polar-opposite phenotypes (e.g., high and low exploratory behavior) and various intermediate forms in between. Importantly, the fitness advantages of contrasting behaviors may differ within various contexts, sometimes dramatically [18]. For example, highly exploratory behavior can be advantageous in conditions of resource competition, but disadvantageous in a higher-predation-pressure environment [18,19,20,21]. Despite the growing number of studies showing high behavioral variability in foraging performance in animals, what could be the fitness consequences of this variation is still not entirely recognized.

The fitness payoff of a behavior under different scenarios is likely to be the main source maintaining the observed variation in foraging strategy [11,18,22]. Under variable conditions, the diversity in behavioral strategies can be maintained if a different performance results in different cost and benefits within different contexts. If so, performance may vary as a function of interactions between social and/or ecological selective forces, which can help reveal the complex interplay of intrinsic and extrinsic factors shaping behavioral variability [23,24,25,26].

Here, we examined the fitness consequences of various behavioral strategies during foraging in wild-ranging hummingbirds. This avian group is known for their extreme metabolism, with a high need for energy intake that makes them constantly motivated to forage [27]. As such, they are expected to be under strong selection for maximizing foraging efficiency, which can have a considerable effect on fitness. This is particularly significant for traplining foraging species, which move around in a route-like fashion across the habitat to visit dispersed flower patches [28,29]. Such a free-ranging strategy entails a period of high vulnerability for foraging individuals, exploited by a wide range of predators [30,31,32,33,34,35], which in turn results in a trade-off between efficient foraging and risk avoidance. The differential payoffs in the selective landscape given by these two factors are expected to shape the foraging strategies of hummingbirds.

In this study, we observed the foraging behavior of the long-billed hermit hummingbird (LBH, *Phaethornis longirostris*). The species is a relatively large-sized hummingbird of humid neo-tropical lowlands. It exhibits a lekking mating system, in which males sing and display from traditional areas inside the forest for mate attraction during an eight-month breeding season [28]. Unlike many hummingbirds, hermits do not defend a patch of flowers, but visit widely scattered flowers in a foraging route (i.e., traplining). Aggressive interactions related to disputes over lek territories are common [29]. Floaters can be found singing from perches of territorial males, while the latter are absent-foraging. Territorial ownership is typically regained after an aggressive interaction with intruders, suggesting a selective pressure to reduce the foraging time and increase territory attendance. Indeed, more efficient foraging males are more likely to own a lek territory [29].

To examine the payoffs of different behavioral strategies under a trade-off between food resource exploitation and risk avoidance, we considered the issue in the context of a low and high level of perceived threat. We also analyzed the issue in regard to three behavioral axes, commonly linked to the exploitation of food resources: (a) exploration (number of foraging spots (i.e., feeders) used during the foraging visit), (b) risk avoidance (latency to approach the foraging spot to forage) and (c) arousal (amount of movements during the foraging event) [1,18,19,36,37]. We hypothesized that foraging efficiency (expressed as the amount of time spent on feeding in respect to the total time of the visit at the feeder) may be lower under threatening conditions. We also hypothesized that foraging efficiency is further modulated by the three behavioral traits (exploration, risk avoidance and arousal), and that high values of these traits can be expected, which will negatively affect foraging efficiency. 

## 2. Methods

### 2.1. Fieldwork

We carried out the study at La Selva Biological Station, Costa Rica (10°23′ N, 84°10′ W) between May and June 2015. We captured and marked 21 individuals with foam tags (total weight of 0.02 g, which is ~0.3% of LBH body mass, 6 g) with unique color combinations, attached to the birds’ back and chest with nontoxic eyelash glue [29]. To evaluate the foraging efficiency of marked individuals and quantify it within low- and high-risk context, we conducted a field experiment using hummingbird feeders. Commercial feeders (Perky model Pet #209B, 900 mL) were modified to have a single opening for accessing “nectar”. Three feeders were arranged in a line (separated by ca. 10 cm distances from each other, Figure 1) and filled with fresh nectar (~30% sugar–water). The set-up was located at a distance of ca. 100 m from the lek border. The feeders (filled with nectar/changed daily) were exposed in the field for two weeks before the onset of experiments, to habituate the birds to feeders. All marked birds were observed on the lek after being caught, either defending territory or as floaters (i.e., all of them survived the capturing and marking procedure, and behaved normally at the lek area). The location of the feeders did not overlap with the foraging area of all the captured individuals; however, for the total, we had 12 visitors at the feeder area.

The experiment session consisted of two consecutive phases performed on the same day, within low- (first phase, control) and high-risk contexts (second phase, experimental). The birds were allowed to forage on the nectar spontaneously, and their behavior was recorded with a commercial camera (continuous recording mode; model: Fujifilm HS30); the camera was set up on a tripod at a distance of ca. 10 m from the feeders (zoomed on the feeder area). During the second phase, we glued a dead bullet ant (*Paraponera clavata*; found dead in the forest) to each of the three feeders (2 cm from the flower hole) to simulate a threat. Although the ant is not a predator of hummingbirds, the presence of large insects on flowers often scares various hummingbird species [37], including long-billed hermits (MAS., pers. obs., Appendix A). Given the existence of behavioral syndrome based on fear generalization, wherein fear of one danger is correlated to the fear of another [38], we assumed that despite not being a danger on its own, through its connotations with truly dangerous insects and its overall novelty, the ant would be perceived by the birds as a threat [37]. Importantly, the attached bullet ants did not completely scare the birds off, allowing for the quantification of their foraging behavior in these circumstances. Hence, attaching the bullet ant to the feeders was expected to resemble situations of increased risk of being injured, while still exploiting the feeding resource.

We performed three complete experiment sessions (with two phases) and two incomplete sessions (with the control phase only due to the weather conditions) all within two weeks (with 2–3 days between the sessions). The two complete (two phases) and two incomplete sessions (control phase only) were conducted in the mornings, when the foraging activity peak occurs [28], and only a single complete experiment session was performed in the afternoon, when the activity is lower. Since the time of the day which we considered in the present study did not affect the foraging efficiency considerably (generalized mixed effect model: foraging efficiency ~ hour (fixed effect; numeric) + birdID (random effect); estimate: −0.009 ± SE 0.005, t = −1.74, df = 137.97, *p* = 0.08), we did not consider it in further analyses. Furthermore, since the entire experiment session was completed within a relatively short window of time (up to three hours), and both phases were completed during the same time window, we do not expect time to bias the comparisons of the control and experimental phases. 

The duration of the control phases, including those from incomplete sessions, varied from 0.5 to 3 h. Owing to the regular visits of the birds under these control conditions (ranging from 1 to 12 per hour per individual), we could record multiple visits of focal individuals, with an average of 6 visits per individual (range 2–18). For the experimental phases, we kept recording until all visitors observed during the preceding control phase returned to the feeders, resulting in an average of 3 visits per individual during this phase (range: 1–7). The control phase was always performed before the experimental one, as doing so, we could ensure the recruitment of individuals at both phases. The presence of the bullet ant on their very first encounter with the feeders in a given day might have precluded the birds to explore the resource. To mitigate the potentially negative effect of bullet ant exposure on the frequency of visits, we performed the sessions with a 2–3 day gap in between. For all these practical reasons, we could not randomize the treatment and control phases. A potentially confounding effect that might bias our results due to the lack of randomization is habituation. Under this scenario, familiarity with the experimental set up results in a foraging efficiency trend over time, regardless of the treatment. To address this concern, we analyzed individual foraging efficiency over consecutive visits using data from control phases (see Appendix A). We found that foraging efficiency changes over time only for one individual, and that happened after a considerable number of visits. Furthermore, in this case, foraging efficiency improved over time. If this pattern had affected the observed difference in efficiency between the treatments, we could expect higher efficiency in the high-risk phase, which was always conducted after the control phase. However, this was not the case (Appendix A). Consequently, we treated all visits of particular individuals as independent data points, but obviously controlling for their identity in further analyses. 

### 2.2. Videos Analysis

We screened the video recordings using VLC software (https://www.videolan.org/vlc/download-windows.html, accessed on 6 June 2015) to locate and cut out video fragments with the foraging visits of focal birds. The events of two or more individuals (long-billed hermits or a different species) visiting the feeders at the same time were uncommon and were excluded from analysis, as interactions disrupted regular foraging behavior. A foraging visit was considered whenever a bird inserted the bill into a feeder at least once. For each bird’s visit, we established key time-points (black circles in Figure 1) with 0.1 s precision, using Cowlog software 2 [39]. Based on these time-points, we calculated the duration of latency to forage, defined as the interval between the appearance in the feeder area (when hovering in front of the feeder at a distance of around 0.5 m was initiated) and the onset of foraging (time from a to b in Figure 1); the duration of the feeding interval/s (time from b to c in Figure 1; on average 4.9 events, with a range of 1–26); duration of the feeding break/s (time from c to d in Figure 1); the duration of the total foraging (time from b to g in Figure 1); and the duration of the total foraging visit (time from a to g in Figure 1). For each visit, we also noted which and how many times each of the three feeders were used by the focal bird.

To quantify bird movements around the feeders, we took advantage of the line arrangement of the feeders that greatly restricted bird activity into two axes. This simplified the subsequent analyses, as operating in a two-dimensional space, we could establish Cartesian coordinates of a bird’s position for each video frame, using the software Tracker version 5.1.5 (https://www.physlets.org/tracker, accessed on 6 June 2015). Based on those coordinates, we calculated the distances between two birds’ positions using the Pythagorean theorem.

### 2.3. Parameters

We calculated foraging efficiency as the ratio of the total duration of foraging (sum of the duration of all feeding intervals) to the total duration of the foraging visit. To characterize the foraging strategies, we measured three behavioral features that have been linked to intrinsic individual characteristics in other species and have been shown to affect foraging: exploration, risk avoidance and arousal [1,18,19,20,36]. As a proxy for exploration, we utilized the rate of the visited feeder: the number of feeders divided by the total duration of the visit, as the absolute number of feeder changes is likely to be a function of the time spent at the feeders. As a proxy for risk avoidance, we used latency to start foraging (the very first use of the feeder during the visit), as defined above (time from a to b in Figure 1). For both parameters, the higher the value, the stronger the exhibited behavior. As arousal, we considered the coefficient of variance in the spatial distances covered by an individual during the whole foraging visit, divided by the number of visited feeders. Birds changing the position frequently (high value of the coefficient) were assumed to exhibit higher arousal.

### 2.4. Data Analysis

All analyses were performed in R [40]. Since all parameters had a skewed distribution, we log-transformed them prior to the analyses (Appendix A). To examine how consistent the birds were in their behavior during the foraging, we estimated the repeatability of all the examined parameters (foraging efficiency, exploration, risk avoidance and arousal) using the rptR package [41]. To do this, we used behavioral features measured during the low-risk (control) treatment only, which represented undisturbed conditions, and for which we recorded multiple visits per individual. When estimating repeatability, we considered behavioral traits as single-response variables and bird identity as a varying intercept factor (i.e., random effect) [41]. 

To assess variation in foraging efficiency (response variable) as a function of the risk level and intrinsic behavioral features (exploration, latency and arousal as predictors), we applied a Bayesian generalized linear mixed model with individuals as a random effect. We fit three models representing alternative hypotheses explaining the variation in foraging efficiency. The first model represents a more traditional view of foraging behavior, in which efficiency is only affected by the level of risk. The second model included an interaction among behavioral traits and the risk level, which represents a more nuanced scenario in which the interplay between the risk level and intrinsic behavioral differences determines foraging efficiency. There was only little collinearity between the predictors (VIF for each of the parameters in the model < 1.5, and the correlation coefficient was also low and ranged from −0.21 to −0.10; Appendix A). We ran this model with a single predictor for intrinsic behavioral traits (separately considering arousal, exploration and risk avoidance), as well as in the form of a global model, with all behavioral predictors included. The two approaches yielded qualitatively similar results; therefore, we presented here only the outcome of the global model, while the outcome of the single-behavior predictor models are presented in Appendix A. Finally, we fit an intercept-only model, representing the scenario in which the proposed predictors do not affect efficiency. The three alternative models were compared using model selection based on the deviance information criteria (DIC) [42]. The models were fit using the R package MCMCglmm [43].

## 3. Results

Repeatability was moderate, but significant for all behavioral traits, except arousal (Figure 2). When predicting foraging efficiency, the model, including all intrinsic behavioral traits and their interaction within the risk predation context, performed significantly better than a simpler model including only the risk context (Table 1). All parameters and their interactions were significant in this model, except for risk avoidance (Table 2). Overall, foraging efficiency was lowered within a higher-risk context (Figure 3) and the effect of intrinsic behavioral features on the foraging efficiency was context-dependent (Table 2). The most dramatic effect was found in respect to exploratory behavior, which was positively related to foraging efficiency within a low-risk context, but the opposite pattern was observed within a higher-risk context (Figure 4). Arousal was positively related to foraging efficiency, and this was particularly pronounced when the birds faced a higher risk (Figure 4). Risk avoidance, overall, tended to decrease foraging efficiency, but that was not significant and did not differ between the risk levels (Figure 4).

## 4. Discussion

As expected according to the risk allocation hypothesis [5,6], the foraging efficiency of long-billed hermits decreased in response to threatening conditions. However, behavioral performance relating exploration, risk avoidance and arousal additionally affected the foraging efficiency, and interestingly, it was affected in a condition-dependent manner. These results suggest that a range of strategies, instead of a single fixed strategy, should be considered when modeling foraging behavior within different contexts. The results also highlight the importance of behavioral variability in shaping the evolution of foraging strategy.

Despite not being a specific prey target, hummingbirds may be opportunistically hunted by a wide range of predators, including insects [31,32,33,34,35]. This imposes a considerable predation risk and favors the evolution of vigilant behavior. Indeed, we found that experimental exposure to an ant, potentially representing just the risk of being bitten (but not eaten), was enough for LBHs to exhibit some behavioral changes. Importantly, despite this risk, the birds did forage, although with a lower efficiency. This places LBHs in a group of species representing the so-called paradox of the risk allocation hypothesis: under conditions of frequent predator presence, prey might need to forage actively, even though a threat (either potential or real predator) is present [5,6,45]. In that context, it is worth to evoke one of the assumptions of the risk allocation hypothesis, which is “living on the edge” in terms of meeting energy demands [5,7]. This assumption seems to be rarely met in most animal species used to test the hypothesis so far [45]. Hummingbirds, given their extreme metabolism rate, could be a rare example when the assumption is actually true.

Under control conditions, individuals exhibiting more exploratory behavior also had an overall higher efficiency during the foraging visit. The opposite pattern was observed for the experimental conditions. A simple reason for this shift could be that the time spent switching feeders was longer under risk conditions, which increased the duration of the visit, probably associated with the need to analyze “de novo” the risk situation. Importantly, individuals were consistent during the time of their exploration, which could indicate that this behavior is related to personality [19,22,46,47]. If the exploratory behavior was indeed a personality trait, and that trait had different fitness consequences in regard to predation, the predation pressure is likely to shape the distribution of exploration phenotypes in the population. Unfortunately, examining bird behavior under limited time and contextual space, we were not able to test this prediction. Nevertheless, to encourage future studies, we highlight the potential role of threats in the environment in the evolution of personality [22,48,49].

There was a clear tendency for risk avoidance behavior to negatively affect the foraging efficiency. In the global model, that we presented in the main text, it was not significant but the effect was apparent in a single-trait model (Appendix A). This indicates that in the most extreme scenario, individuals exhibiting high-risk aversion may tend to jeopardize their survival in terms of energy intake, while individuals with low-risk aversion, although benefiting from high foraging efficiency, would be more likely to be predated. If the risk avoidance was related to the bird personality, the relationship between that and foraging efficiency would contribute to the selection of given behavioral phenotype at a given predation risk level. Consistently, frequent changes and/or an unpredictable level of risk predation in the environment would maintain variability in this behavioral phenotype [18]. Again, we cannot boldly make conclusions about bird personality here, but the results are intriguing and encourage exploring this research avenue in the future.

To maximize fitness, hummingbirds should adaptively allocate both the exploratory and risk avoidance behaviors. Here, we considered the issue in respect to a given species, but interpreting our study in a broader context, we could speculate that exploratory and risk avoidance should be differently allocated in hummingbirds that differ in foraging strategy, such as trapliners and territorials. The two groups are likely to experience different risk levels of predation, and so behaviors such as exploration and risk avoidance could also differ. Weighting the relative effect of these intrinsic behavioral features can allow us to understand the evolutionary factors shaping foraging performance more accurately [50,51,52]. 

An increasing foraging efficiency with an increase in arousal may be counterintuitive at first glance, as the time allocated to movements potentially limits the time for foraging. However, arousal was not a repeatable trait; thus, birds’ arousal may simply reflect their nutritional state, and may vary considerably between different days and even visits. In such case, more active individuals could be more effective during the foraging, owing to their good body condition or high motivation to forage. 

Both the exploratory and risk avoidance behaviors were quite repeatable for individuals. Although more studies are needed to properly examine how stable this repeatability is within different contexts and over a longer period, our results suggest that these two behaviors could be related to bird personality [22,53,54]. In a constantly changing environment, the varying fitness consequences of a given behavioral phenotype would maintain the variation in animal personality [18]. If indeed the exploratory and risk avoidance behavior are at least partially heritable traits, this can be expected to play an important role in the evolution of behavioral phenotypes under diverse conditions of predation pressure. Hummingbirds stand as a useful model for the study of animal behavioral syndromes and their interaction with critical natural history features in the wild.

## 5. Conclusions

Although with a lower efficiency, long-billed hermits foraged despite an elevated level of risk in the environment. This makes them a good model species for studies on the risk allocation hypothesis, where the propensity to forage is measured in the context of predator presence. Our results show not only that the foraging efficiency of an individual is affected by the presence of a threat, but also that the efficiency depends on the behavioral performance of the individual. More exploration was associated with a higher foraging efficiency under no-risk conditions, but it was lower when the birds were exposed to a risk. Regardless of the conditions, arousal was positively associated with foraging efficiency, while risk avoidance was related negatively. Importantly, exploratory behavior and risk avoidance were quite repeatable behaviors, suggesting that they may be related to the intrinsic traits of individuals. Our findings highlight the importance of taking into account additional behavioral dimensions to better understand the foraging strategies of individuals.

## Figures and Tables

**Figure 1 animals-13-01997-f001:**
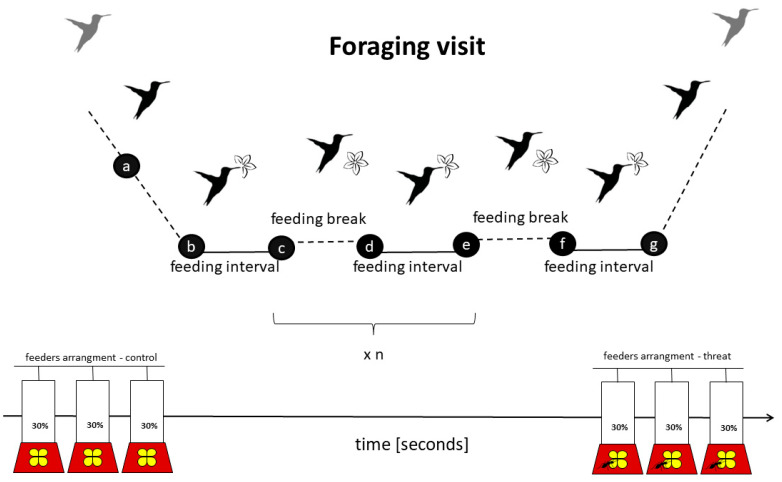
Scheme of the foraging visit—the total time spent by focal birds at the feeder area, with at least one feeding event. Time-points crucial for data analysis are denoted with black circles and labelled with letters to denote particular events relating the components of the foraging visit: (a) onset of the foraging visit (appearance in the feeder area, usually hovering in front of the feeder); (b, d, f) onsets of consecutive feeding events (i.e., inserting the bill into the flower hole of the feeder); (c, e) end of the respective feeding events (i.e., removal of the bill from the flower hole of the feeder); (g) end of the foraging visit (i.e., the end of the very last feeding event during the foraging visit). Multiple feeding intervals were possible (1–26, mean: 4.9).

**Figure 2 animals-13-01997-f002:**
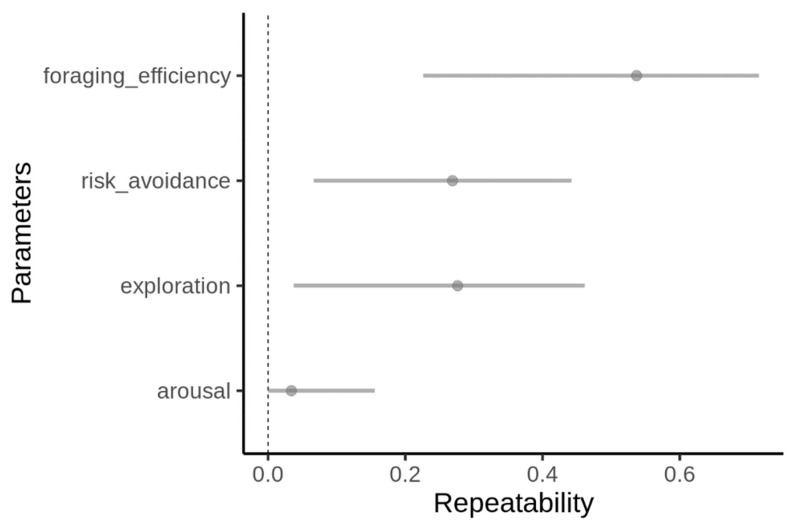
Repeatability estimates of foraging efficiency and behavioral parameters in the long-billed hermit measured under control conditions, analyzed using a linear mixed-effects models [44]. Dots represent the value of the repeatability coefficient, and the bars 95% confidence intervals.

**Figure 3 animals-13-01997-f003:**
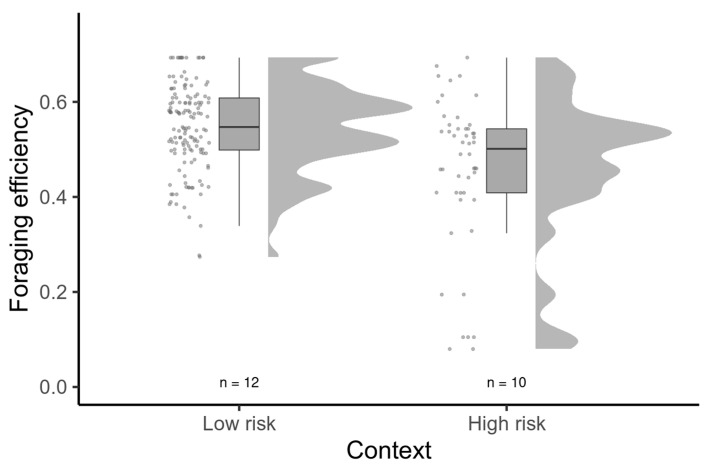
Foraging efficiency in the context of low and high levels of perceived risk. Data points, boxplot (with the median and inter-quartile range) and data distribution are presented.

**Figure 4 animals-13-01997-f004:**
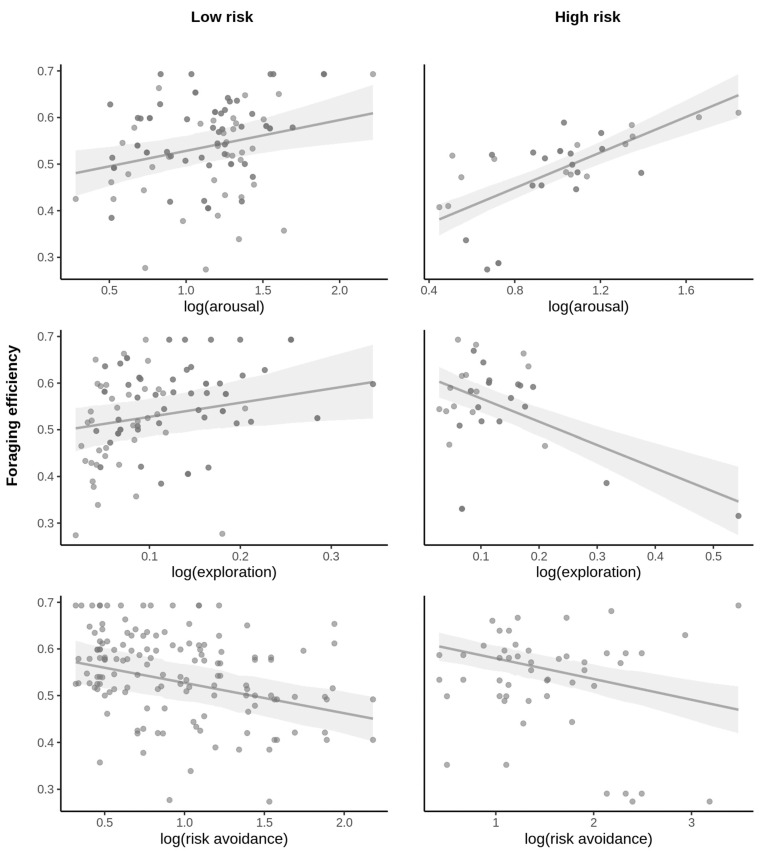
Foraging efficiency of long-billed hermits relating to their behavioral performance (exploration, risk avoidance and arousal) in the context of low and high levels of perceived risk of predation. Regression lines presented with 95% confidence intervals (shadow area).

**Table 1 animals-13-01997-t001:** Ranking of the models explaining the foraging efficiency of long-billed hermits, ordered by delta Deviance Information Criterion (DIC; Akaike’s Information Criterion AIC yields the same conclusions).

Predictors	df	DIC	ΔDIC	Weight DIC	AIC	ΔAIC	Weight AIC
md_all_interactions	10	−400.0909	0.00	1	−396.3073	0.00	0.99
md_arousal_exploration	8	−388.2385	11.85	0	−386.2831	10.02	0.01
md_arousal_risk_avoidance	8	−378.9807	21.11	0	−376.8184	19.49	0.00
md_arousal	6	−363.3410	36.75	0	−363.2509	33.06	0.00
md_risk_avoidance_exploration	8	−350.1568	49.93	0	−348.8140	47.49	0.00
md_exploration	6	−345.7716	54.32	0	−346.4065	49.90	0.00
md_risk_avoidance	6	−315.2258	84.87	0	−315.0929	81.21	0.00
md_context	4	−308.6036	91.49	0	−310.7995	85.51	0.00
md_null	3	−296.3098	103.78	0	−299.8347	96.47	0.00

**Table 2 animals-13-01997-t002:** Effects of behavioral variables and experimental context (low/high risk) on the foraging efficiency of long-billed hermits (N individuals = 12, N data points = 192). Common intercept = 0.4548, effects are the slope estimates derived from the first top Bayesian MCMC generalized linear model (Table 1). Significant effects are denoted in bold.

Predictor	Effect Size	CI 2.5%	CI 97.5%	pMCMC
contextHigh risk	−0.1409	−0.2732	−0.0132	0.0322
Arousal	0.0684	0.0275	0.1083	0.0006
Exploration	0.3686	0.1244	0.6167	0.0023
risk_avoidance	−0.0327	−0.0663	0.0023	0.0641
contextHigh risk:arousal	0.2445	0.1541	0.3436	0.0001
contextHigh risk:exploration	−0.8355	−1.1641	−0.4925	0.0001
contextHigh risk:risk_avoidance	−0.0270	−0.0793	0.021	0.2918

## Data Availability

Data and data analysis scripts associated with the manuscript are included as Appendix A.

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
