# Peer review of "Foraging, Fear and Behavioral Variation in a Traplining Hummingbird"

_animals, 2023, doi:10.3390/ani13121997_

Round 1
Reviewer 1 Report
An interesting topic well presented. This paper aims to explore variation in foraging efficiency of LBHs with regard to exploration, risk avoidance and arousal. It's contributions to science take into account behavioural dimensions to better understand foraging strategies of individuals.
The manuscript is clear although minor alterations are needed in the English language used, commendable however if English is not authors first language. It is presented in a well structured manner, however does not include a Conclusions section and it is recommended that some of the methods section are included in different parts of the manuscript. Good use of references (although included in full in manuscript not as bracketed numbers). Methods are scientifically sound however as someone naïve to some of the methodology used a little further clarification would be useful for repeatability of methods. Figure 3 would benefit from further description to explain what it is showing. Good to see the reporting of effect sizes in table 2. Can the discussion/conclusions draw clearer meaning from the results and relate back directly to figures/model outcomes? The discussion section would benefit from a paragraph on the limitations of the research and recommendations for future research based on the results found. A "simple summary" may also be useful for brief science communication. Additional details included on attached reviewed manuscript.

The manuscript is clear although minor alterations are needed in the English language used, commendable however if English is not authors first language.
Author Response
Dear Reviewer,
We carfully considered them all, found them really helpful and modified the manuscript accordingly. We hope the changes we made are satisfactory. Thank you very much for your time and constructive review.
REVIEWER comments:
An interesting topic well presented. This paper aims to explore variation in foraging efficiency of LBHs with regard to exploration, risk avoidance and arousal. It's contributions to science take into account behavioural dimensions to better understand foraging strategies of individuals.
The manuscript is clear although minor alterations are needed in the English language used, commendable however if English is not authors first language.
REPLY: We carefully proof-read the manuscript and correct all the spellings, we hope the changes are satisfactory.
It is presented in a well structured manner, however does not include a Conclusions section and it is recommended that some of the methods section are included in different parts of the manuscript.
REPLY: We added a paragraph with conclusions (lines: 842-853) and moved part of the methods section into the introduction as suggested (now 96-106). For the part on analyses and their output, performed for the data quality check, which we placed in the methods and Reviewer recommended to move to the results section, we would argue that methods section is actually more suitable place for that content. This is because these analyses are not the core of the study and we performed them to justify some other analytical procedures. As such we believe they fit well in the methods section (see the lines: 479-500)
Good use of references (although included in full in manuscript not as bracketed numbers).
REPLY: We adjusted the reference style to the Animals requirements.
Methods are scientifically sound however as someone naïve to some of the methodology used a little further clarification would be useful for repeatability of methods.
REPLY: Clarifications made throughout the whole text, especially in the places pointed out by the Reviewer on the pdf document.
Figure 3 would benefit from further description to explain what it is showing.
REPLY: The figure was actually improved (data points/boxplot for the median and inter-quartile values/data distribution are now presented in separate forms). Then, description is elaborated: “Foraging efficiency in the context of low and high levels of perceived risk of predation. Data points, boxplot (with median and interquartile range) and data distribution are presented.” Also description for other figures were elaborated.
Good to see the reporting of effect sizes in table 2. Can the discussion/conclusions draw clearer meaning from the results and relate back directly to figures/model outcomes?
REPLY: We modified the text of the discussion to make it clearer. We hope the changes are satisfactory.
The discussion section would benefit from a paragraph on the limitations of the research and recommendations for future research based on the results found.
REPLY: We added some sentences about the limitations of the study and suggestion of future research (lines: 472-500, 710-713, 724-276, 829-831, 835-838)
[Lines 472-500: “The fixed sequence of the experiment phases could bias results on foraging efficiency if that would decrease over the time of experiment session (e.g. due to satiation effect over the whole experiment duration). To address this concern, we analyzed foraging efficiency over the consecutive visits using data from control phases (see Supplementary Online Materials). We found that although foraging efficiency may change over the time, that apparently happens only after a considerable number of visits of an individual. Since in our data set, considerable number of visit was the issue with single individuals at single control phases, we considered that issue of not particular importance. Besides, foraging efficiency improved over the time, and given the fact that experiments with bullet ants were always performed after the control phases, if the fixed sequence of the phases biased the results, we would observe an increase or no difference, instead of decrease in foraging efficiency. Since that did not happen, the results of the experiment are apparently solid (Supplementary Online Materials: Fig. S1 and S2).”]
[Lines 710-713: “Unfortunately, examining bird behaviour in limited time and contextual space we are not able to test this prediction currently. Nevertheless, to encourage future studies, we highlight potential role of threats in the environment in the evolution of personality”]
[Lines 724-726: “Again, we cannot boldly conclude here about birds personality but the results are intriguing and encourage to explore this research avenue in the future.”]
[Lines 829-831: “Although more study are needed to properly examine how stable this repeatability is over different contexts and a longer period, our results suggest that these two behaviors could be related to birds personality.”]
[Lines 835-838: “We are currently not able to perform any analysis of that kind given the relatively low number of tested individuals and short study period but we point out hummingbirds as potential animal model species in the studies of animals personality.”]
A "simple summary" may also be useful for brief science communication. Additional details included on attached reviewed manuscript.
REPLY: We added conclusion section as required into the body text (lines: 842-853)
- The text fragment – description of the study species, which was in methods section, was moved into the introduction, as suggested by Reviewer.
- The description of the experimental design (ant attachment) was clarified (lines 443-445), and Figure 1 was also improved to address this concern:
[Lines 443-445: “During the second phase, to simulate a threat we glued a dead bullet ant (Paraponera clavata; found dead in the forest) to each of the three feeders, 2 cm from the flower-hole.”]
- We kindly disagree about the fragment about analyses and their output on the effect of the experimental design – we put it into the methods section (lines 479-500), and Reviewer suggested to move it to the results. Since these analyses are not the core of the study, they were performed to justify the experimental design, we believe they should be presented in the methods section (and supplementary materials); all this is also explained above.
- Reviewer suggested to add ethical details into the methods section but this information is provided in the special section, following the journal requirement (Institutional Review Board Statement: lines 911-917). Thus we did not change that. However, relevant information about the birds fate after capturing is also provided in the methods (lines 318-320)
[Lines 318-320: “All marked birds were observed on the lek after being caught, either defending territory or as floaters (i.e. all of them survived the capturing and marking procedure and behaved normally at the lek area”]
[Lines 911-917: “All activities (birds marking, feeding, and video recording/observations) were performed with the greatest care. The capturing and marking procedure did not have apparent effect on birds survival, as all the individuals where observed in the lek and/or at the feeders area after the capturing. The foam with the birds were marked was of negligible weight (0.02 g, which constitutes ~0.3% of average body mass (6g) of LBH). The study was reviewed and authorized by the Costa Rican Ministerio del Ambiente y Energia (063-2011-SINAC), and performed in accordance with their guidelines and regulations.”]
- Limitations of the study and suggested future research are discussed in the lines: 472-500, 710-713, 724-276, 829-831, 835-838); already addressed in the letter (please see above).
- All the typos pointed out on the pdf file were fixed, and headings for the figures were clarified/elaborated.
Reviewer 2 Report
In this experiment, the authors investigated how individually marked hummingbirds exploited artificial resources. The main idea was to determine whether some traits associated with food exploitation were repeatable and then to assess whether these traits affected foraging efficiency in a similar fashion in different treatments. I agree that this sort of experiment can be useful to examine how predation risk can shape resource exploitation. It was also nice to see a tropical species used as subjects for this sort of experiment.
My main concern is that the ant used for the higher risk treatment is not really a threat of predation. The predictions made by the authors relate to predation risk. I am not even sure the hummingbirds perceived the ant, which was dead, as a biting threat at all. Perhaps it was viewed as a novel object on the feeder and what was measured here was a reaction to novelty. Granted, novelty can be perceived as a threat but one has to be careful about how the treatments are labeled and described. I have more minor issues listed below.
Line 85: Typo for ‘diferent’
Line 95: Please be a little more explicit about how each of the axes is expected to affect foraging efficiency.
Line 188: Why not simply consider id as well as visit number in the analyses? This would control for any possible effect of visit numbers on foraging efficiency.
Line 221: Just to be clear, avoidance was only measured for the first flower in a visit, yes?
Line 223: I am not sure I understand why you used CV rather than total distance covered during a visit. A high CV would suggest more variability in distances moved during a visit rather than more distances covered. Please elaborate on this choice. Also, I am not clear about the meaning of the term arousal in this context. Why would more arousal translate into more movements?
Line 245: Correlation coefficients are odd as a choice to assess collinearity. There are tools in R to assess multicollinearity with inflation factors.
Figure 3: Please provide more explanation about the various items present in this figure (the dot, the shaded area, etc.).
Figure 4: Define the regression line and shaded area.
Line 287: Any model including food reward and predation risk would make these predictions. The Lima and Bednekoff model has very particular constraints not applicable here. A more general reference is needed such as Houston, A. I., McNamara, J. M., & Hutchinson, J. M. C. (1993). General results concerning the trade-off between gaining energy and avoiding predation. Philosophical Transactions of the Royal Society London Series B, 341, 375-397.
Line 317: Consistency was only measured over a few weeks. Is this considered sufficient to imply personality?
Line 342: To really invoke predation pressure, one would need an experimental treatment involving a threat of predation I believe.
Line 346: I am still not clear what arousal was actually measuring and perhaps this explains why it was not repeatable.
There are several typos and weak sentences in here but the paper was readable.
Author Response
REVIEWER 2
Dear Reviewer,
Thank you for your time and constructive review. We considered carefully all the comments, found them all helpfull and accordingly we modified the manuscript. We hope the changes are satisfactory.
Reviewer comments:
In this experiment, the authors investigated how individually marked hummingbirds exploited artificial resources. The main idea was to determine whether some traits associated with food exploitation were repeatable and then to assess whether these traits affected foraging efficiency in a similar fashion in different treatments. I agree that this sort of experiment can be useful to examine how predation risk can shape resource exploitation. It was also nice to see a tropical species used as subjects for this sort of experiment.
My main concern is that the ant used for the higher risk treatment is not really a threat of predation. The predictions made by the authors relate to predation risk. I am not even sure the hummingbirds perceived the ant, which was dead, as a biting threat at all. Perhaps it was viewed as a novel object on the feeder and what was measured here was a reaction to novelty. Granted, novelty can be perceived as a threat but one has to be careful about how the treatments are labeled and described.
REPLY: We agree that ant can’t be treated as a predator model. However, we assume that it can be perceived as a threat (based on the fear generalization as recently highlighted by Sih et al 2023), and our results demonstrate it is the case. Then we consider the issue and interpret birds behaviour in a broader context – predation pressure – which is very much relevant. To address the concern, however, we tuned down the whole text about the predation, both in the methods and the discussion (lines 443-451, 731-735).
[Lines 731-735: “The two groups are likely to experience different threat of predation, and so behaviors like exploration and risk avoidance could also differ. All that might then affect foraging performance [48–50]. Thus, our study suggest possible importance of perceived threat in shaping foraging strategy of various hummingbirds.” ]
[Lines 443-451: “During the second phase, to simulate a threat we glued a dead bullet ant (Paraponera clavata; found dead in the forest) to each of the three feeders, 2 cm from the flower-hole. Although the ant is not a predator of hummingbirds, presence of large insects on flowers often scares various hummingbird species (e.g. Carr and Golinski 2020), including Long-billed Hermits (MAS., pers. obs, Supplementary Online Materials: videos 1 and 2). Given existence of behavioral syndrome based on fear generalization , wherein fear of one danger is correlated to fear of other [38] we assumed that the anteven if not being a danger on its own, by its connotations with truly dangerous insects and its overall novelty would be perceived by birds as a theat. Importantly, attached bullet ants did not completely scare birds off, allowing the quantification of their foraging behaviour in these circumstances. Hence attaching the bullet ant to the feeders was expected to resemble situations of increased risk of being injured while still exploiting the feeding resource.”]
I have more minor issues listed below.
Line 85: Typo for ‘diferent’
REPLY: Fixed.
Line 95: Please be a little more explicit about how each of the axes is expected to affect foraging efficiency.
REPLY: We elaborated the issue (lines, 117-303): “Then we hypothesized that foraging efficiency is further modulated by the three behavioral traits (exploration, risk avoidance and arousal), expecting that high level of each of the trait negatively affects the foraging efficiency (i.e. frequent changes of the foraging spot, long latency to approach the feeder and lot of movement around the feeder all it may limit the time for feeding). “
Line 188: Why not simply consider id as well as visit number in the analyses? This would control for any possible effect of visit numbers on foraging efficiency.
REPLY: We did considered birds id in the analyses but still the consecutive visits of the same individual are not entirely independent (for example bird exposed to a treat at first visit may behave differently at the very second visit; for this reason we focused on control phases only while examining repeatability of behaviour). However for simplicity of the analyses we considered birds visits as independent data points, and to justified this approach we performed additional analyses (lines: 417-502)
[Lines 417-502: “To address this concern, we analyzed foraging efficiency over the consecutive visits using data from control phases (see Supplementary Online Materials). We found that although foraging efficiency may change over the time, that apparently happens only after a considerable number of visits of an individual. Since in our data set, considerable number of visit was the issue with single individuals at single control phases, we considered that issue of not particular importance. Besides, foraging efficiency improved over the time, and given the fact that experiments with bullet ants were always performed after the control phases, if the fixed sequence of the phases biased the results, we would observe an increase or no difference, instead of decrease in foraging efficiency. Since that did not happen, the results of the experiment are apparently solid (Supplementary Online Materials: Fig. S1 and S2). Consequently, we treated all the visits of particular individuals as independent data points but obviously controlling for their identity in further analyses”].
Line 221: Just to be clear, avoidance was only measured for the first flower in a visit, yes?
REPLY: Yes, we clarified it in the text (lines 533-535): “As a proxy for risk avoidance we used latency to start to forage (the very first use of the feeder during the visit); as defined above (time from a to b on Fig. 1)”
Line 223: I am not sure I understand why you used CV rather than total distance covered during a visit. A high CV would suggest more variability in distances moved during a visit rather than more distances covered. Please elaborate on this choice. Also, I am not clear about the meaning of the term arousal in this context. Why would more arousal translate into more movements?
REPLY: We calculated the coefficient of variance to account changes of the location of a bird, which we believe is the most relevant in the considered context. The higher variation is in that (so high cv) the more “visible” individual could be; its position is less predictable. For instance for two individuals of the same total distance covered cv may be different, and for that which has cv of higher value we could see its behaviour to be more variable than for the individual of low cv value. We clarified the issue in the text, please see the lines 536-539: “As arousal we considered the coefficient of variance in spatial distances covered by an individual during the whole foraging visit, divided by the number of visited feeders. Birds changing the position frequently (of high value of the coefficient) were assumed to exhibit higher arousal.”
Line 245: Correlation coefficients are odd as a choice to assess collinearity. There are tools in R to assess multicollinearity with inflation factors.
REPLY: We calculated VIF and for all the parameters it was < 2. We added the information into the text (please see the lines: 606-608: “There was only little collinearity between predictors (VIF for each of the parameters in the model <1.5 also correlation coefficient ranged from -0.21 to -0.10; Supplementary Online Materials: Fig. S3).”)
Figure 3: Please provide more explanation about the various items present in this figure (the dot, the shaded area, etc.).
REPLY: Headings for all the figures (including the Fig 3) were clarified and elaborated as suggested. The figure was actually improved (data points/boxplot for the median and inter-quartile values/data distribution are now presented in separate forms).
Figure 4: Define the regression line and shaded area.
REPLY: Headings for all the figures (including the Fig 4) were clarified and elaborated as suggested.
Line 287: Any model including food reward and predation risk would make these predictions. The Lima and Bednekoff model has very particular constraints not applicable here. A more general reference is needed such as Houston, A. I., McNamara, J. M., & Hutchinson, J. M. C. (1993). General results concerning the trade-off between gaining energy and avoiding predation. Philosophical Transactions of the Royal Society London Series B, 341, 375-397.
REPLY: Thank you for the suggestion, we added/replace it wherever it was relevant.
Line 317: Consistency was only measured over a few weeks. Is this considered sufficient to imply personality?
REPLY: We agree that to properly measure personality longer period of time would be recommended but observed repeatability is already promising, thus we evoke this issue and speculate about personality. To address the concern we tuned down relevant text (please see the lines: 705-712, 724-726, 828-831).
[Lines 705-712: “Importantly, individuals were consistent over the time in their exploration which could indicate that this behaviour is related to personality [19,22,44,45]. If the exploratory behaviour was indeed a personality trait, and that trait had different fitness consequences in regard to predation, the predation pressure is likely to shape distribution of exploration phenotypes in the population. Unfortunately, examining bird behaviour in limited time and contextual space we are not able to test this prediction currently. Nevertheless, to encourage future studies, we highlight potential role of threats in the environment in the evolution of personality [22,46,47].”]
[Lines 724-726: “Again, we cannot boldly conclude here about birds personality but the results are intriguing and encourage to explore this research avenue in the future.”]
[Lines 828-831: “Both exploratory and risk avoidance behaviour were quite repeatable for individuals. Although more study are needed to properly examine how stable this repeatability is over different contexts and a longer period, our results suggest that these two behaviors could be related to birds personality [22,51,52].”]
[Lines 833-839; “If indeed the exploratory and risk avoidance behaviour are at least partially heritable traits, one could use them to model an evolutionary scenario for given behavioral phenotypes in various conditions of predation pressure. We are currently not able to perform any analysis of that kind given the relatively low number of tested individuals and short study period but we point out hummingbirds as potential animal model species in the studies of animals personality”]
Line 342: To really invoke predation pressure, one would need an experimental treatment involving a threat of predation I believe.
REPLY: We agree, and as for personality issue, we still discuss the issue in the context of predation as this is relevant but we tuned down the text (please see the lines: 736-730, 782-784).
[Lines 736-730: “Despite not being a specific prey target, hummingbirds may be opportunistically hunted by a wide range of predators, including insects [31,33–35,43]. That imposes a considerable predation risk and favors evolution of vigilance behaviour. Indeed, we found that experimental exposure of an ant, potentially representing just a threat of being bitten (but not eaten), was enough for LBHs to exhibit some behavioral changes.]”
[Lines 782-784: “All that might then affect foraging performance [48–50]. Thus, our study suggest possible importance of perceived threat in shaping foraging strategy of various hummingbirds.”]
Line 346: I am still not clear what arousal was actually measuring and perhaps this explains why it was not repeatable.
REPLY: We calculated the coefficient of variance (not really the total distance covered) to measure the birds overall activity - how frequently and how much it changes its position, as this is what matters the most in the context of potential presence of the predator. However, this behavior is also likely to be affected quite much by birds motivation/nutritional state and that is probably the reason why it was not found to be repeatable. We discuss it in the lines: 785-781, and modified the text to clarify it (lines 586-589).
[Lines 586-589: “As arousal we considered the coefficient of variance in spatial distances covered by an individual during the whole foraging visit, divided by the number of visited feeders. Birds changing the position frequently (of high value of the coefficient) were assumed to exhibit higher arousal.”]
[Lines 785-781: “An increasing foraging efficiency with an increase in arousal may be counterintuitive at first glance, as time allocated to movements potentially limits the time for foraging. However, arousal was not a repeatable trait, thus bird’s arousal may simply reflect its nutritional state, thus it may vary considerably. Then, more active individuals could be more effective during the foraging, owing to their good body condition or high motivation to forage.”]
There are several typos and weak sentences in here but the paper was readable.
REPLY: We carefully proof-read the whole manuscript, removed all the typos and corrected awkward sentences. We hope the changes are satisfactory.